# Multisecond ligand dissociation dynamics from atomistic simulations

Steffen Wolf [1✉], Benjamin Lickert [1], Simon Bray [1,2] & Gerhard Stock [1✉]

Coarse-graining of fully atomistic molecular dynamics simulations is a long-standing goal in order to allow the description of processes occurring on biologically relevant timescales. For example, the prediction of pathways, rates and rate-limiting steps in protein-ligand unbinding is crucial for modern drug discovery. To achieve the enhanced sampling, we perform dissipation-corrected targeted molecular dynamics simulations, which yield free energy and friction profiles of molecular processes under consideration. Subsequently, we use these fields to perform temperature-boosted Langevin simulations which account for the desired kinetics occurring on multisecond timescales and beyond. Adopting the dissociation of solvated sodium chloride, trypsin-benzamidine and Hsp90-inhibitor protein-ligand complexes as test problems, we reproduce rates from molecular dynamics simulation and experiments within a factor of 2–20, and dissociation constants within a factor of 1–4. Analysis of friction profiles reveals that binding and unbinding dynamics are mediated by changes of the surrounding hydration shells in all investigated systems.

[1] Biomolecular Dynamics, Institute of Physics, Albert Ludwigs University, Hermann-Herder-Strasse 3, 79104 Freiburg, Germany. [2] Present address: Bioinformatics Group, Department of Computer Science, Albert Ludwigs University, Georges-Koehler-Allee 106, 79110 Freiburg, Germany. ✉email: steffen.wolf@physik.uni-freiburg.de; stock@physik.uni-freiburg.de

Classical molecular dynamics (MD) simulations in principle allow us to describe biomolecular processes in atomistic detail[1]. Prime examples include the study of protein complex formation[2] and protein–ligand binding and unbinding[3,4], which constitute key steps in biomolecular function. Apart from structural analysis, the prediction of kinetic properties has recently become of interest, since optimized ligand binding and unbinding kinetics have been linked to an improved drug efficacy[5–9]. Since these processes typically occur on timescales from milliseconds to hours, however, they are out of reach for unbiased all-atom MD simulations which currently reach microsecond timescales. To account for rare biomolecular processes, a number of enhanced sampling techniques[10–18] have been proposed. These approaches all entail the application of a bias to the system in order to enforce motion along a usually one-dimensional reaction coordinate $x$, such as the protein–ligand distance.

While the majority of the above methods focuses on the calculation of the stationary free energy profile $\Delta G(x)$, several approaches have recently been suggested that combine enhanced sampling with a reconstruction of the dynamics of the process[19–21]. In this vein, we recently proposed dissipation-corrected targeted MD (dcTMD), which exerts a pulling force on the system along reaction coordinate $x$ via a moving distance constraint[22]. By combining a Langevin equation analysis with a cumulant expansion of Jarzynski's equality[23], dcTMD yields both $\Delta G(x)$ and the friction field $\Gamma(x)$. Reflecting interactions with degrees of freedom orthogonal to those which define the free energy, the friction accounts for the dynamical aspects of the considered process. In this work, we go one step further and use $\Delta G(x)$ and $\Gamma(x)$ to run Langevin simulations, which describe the coarse-grained dynamics along the reaction coordinate and reveal timescales and mechanisms of the considered process. Moreover, we introduce the concept of "temperature boosting" of the Langevin equation, which allows us to speed up the calculations by several orders of magnitude in order to reach biologically relevant timescales.

## Results

**Dissipation-corrected targeted molecular dynamics**. To set the stage, we briefly review the working equations of dcTMD derived in[22]. TMD as developed by Schlitter et al.[24] uses a constraint force $f_c$ that results in a moving distance constraint $x = x_0 + v_c t$ with a constant velocity $v_c$. The main assumption underlying dcTMD is that this nonequilibrium process can be described by a memory-free Langevin equation[1],

$$m\ddot{x}(t) = -\frac{dG}{dx} - \Gamma(x)\dot{x} + \sqrt{2k_B T\Gamma(x)}\,\xi(t) + f_c(t), \quad (1)$$

which contains the Newtonian force $-dG/dx$, the friction force $-\Gamma(x)\dot{x}$, as well as a stochastic force with white noise $\xi(t)$, that is assumed to be of zero mean, $\langle\xi\rangle = 0$, delta-correlated, $\langle\xi(t)\xi(t')\rangle = \delta(t - t')$, and Gaussian distributed. Since the constraint force $f_c$ imposes a constant velocity on the system ($\dot{x} = v_c$), the total force $m\ddot{x}$ vanishes. Performing an ensemble average $\langle\ldots\rangle$ of Eq. (1) over many TMD runs, we thus obtain the relation[22]

$$\Delta G(x) = \langle W(x)\rangle - v_c \int_{x_0}^{x} \Gamma(x')\,dx'. \quad (2)$$

Here the first term $\langle W(x)\rangle = \int_{x_0}^{x}\langle f_c(x')\rangle\,dx'$ represents the averaged external work performed on the system, and the second term corresponds to the dissipated work $W_{diss}(x)$ of the process expressed in terms of the friction $\Gamma(x)$.

While the friction in principle can be calculated in various ways[25,26], it proves advantageous to invoke Jarzynski's identity[23],

$e^{-\Delta G(x)/k_B T} = \langle e^{-W(x)/k_B T}\rangle$, which allows us to calculate $\Gamma(x)$ directly from TMD simulations. To circumvent convergence problems associated with the above exponential average[27], we perform a second-order cumulant expansion which gives Eq. (2) with $W_{diss}(x) = \langle\delta W^2(x)\rangle/k_B T$. Expressing work fluctuations $\delta W$ in terms of the fluctuating force $\delta f_c$, we obtain for the friction[22]

$$\Gamma(x) = \frac{1}{k_B T}\int_{t_0}^{t(x)}\langle\delta f_c(t)\delta f_c(t')\rangle dt', \quad (3)$$

which is readily evaluated directly from the TMD simulations.

As discussed in ref. [22], the derivation of Langevin Eq. (1) assumes that the pulling speed $v_c$ is slow compared to the timescale of the bath fluctuations, such that the effect of $f_c$ can be considered as a slow adiabatic change[28]. This means that the free energy Eq. (2) and the friction Eq. (3) determined by the nonequilibrium TMD simulations correspond to their equilibrium results. As a consequence, we can use $\Delta G(x)$ and $\Gamma(x)$ to describe the unbiased motion of the system via Langevin Eq. (1) for $f_c = 0$. Numerical propagation of the unbiased Langevin equation then accounts for the coarse-grained dynamics of the system. In this way, calculations of $\Delta G(x)$ and $\Gamma(x)$ as well as dynamical calculations are based on the same theoretical footing (i.e., the Langevin equation), and are therefore expected to yield a consistent estimation of the timescales of the considered process. Moreover, the exact solution of the Langevin equation allows us to directly use the computed fields $\Delta G(x)$ and $\Gamma(x)$ and thus to avoid further approximations[29].

The theory developed above rests on two main assumptions. For one, we have assumed that the Langevin Eq. (1) provides an appropriate description of nonequilibrium TMD simulations, and applies as well to the unbiased motion ($f_c = 0$) of the system. This means that, due to a timescale separation of slow pulling speed and fast bath fluctuations, the constraint force $f_c$ enters this equation merely as an additive term. Secondly, to ensure rapid convergence of Jarzynski's identity, we have invoked a cumulant expansion to derive the friction coefficient in Eq. (3), which is valid under the assumption that the distribution of the work is Gaussian within the ensemble. While this assumption may break down if the system of interest follows multiple reaction paths, we have recently shown that we can systematically perform a separation of dcTMD trajectories according to pathways by a nonequilibrium principal component analysis of protein–ligand contacts[30]. This approach bears similarities with the work of Tiwary et al. for the construction of path collective variables[31]. Alternatively, path separation can be based on geometric distances between individual trajectories, making use of the NeighborNet algorithm[32]. Details on the convergence of the free energy and friction estimates, the path separation, and the choice of the pulling velocity are given in the Supplementary Methods and in Supplementary Figs. 1–4.

***T-boosting***. The speed-up of Langevin Eq. (1) compared to an unbiased all-atom MD simulation is due to the drastic coarse graining of the Langevin model (one instead of $3N$ degrees of freedom, $N$ being the number of all atoms). Since the numerical integration of the Langevin equation typically requires a time step of a few femtoseconds (see Supplementary Table 1), however, we still need to propagate Eq. (1) for $\gtrsim 100 \times 10^{15}$ steps to sufficiently sample a process occurring on a timescale of seconds, which is prohibitive for standard computing resources.

As a further way to speed up calculations, we note that the temperature $T$ enters Eq. (1) via the stochastic force, indicating that temperature is the driving force of the Langevin dynamics. That is, when we consider a process described by a transition rate

$k$ and increase the temperature from $T_1$ to $T_2$, the corresponding rates $k_1$ and $k_2$ are related by the Kramers-type expression[29]

$$k_2 = k_1 e^{-\Delta G^{\neq}(\beta_2 - \beta_1)}, \qquad (4)$$

where $\Delta G^{\neq}$ denotes the transition state energy and $\beta_i = 1/k_B T_i$ is the inverse temperature. Hence, by increasing the temperature we also increase the number $n$ of observed transition events according to $n_2/n_1 = k_2/k_1$.

To exploit this relationship for dcTMD, we proceed as follows. First we employ dcTMD to calculate the Langevin fields $\Delta G(x)$ and $\Gamma(x)$ at a temperature of interest $T_1$. Using these fields, we then run a Langevin simulation at some higher temperature $T_2$, which results in an increased transition rate $k_2$ and number of events $n_2$. In particular, we choose a temperature high enough to sample a sufficient number of events ($N \gtrsim 100$) for some given simulation length. In the final step, we use Eq. (4) to calculate the transition rate $k_1$ at the desired temperature $T_1$.

As Eq. (4) arises as a consequence[29] of Langevin Eq. (1), the above described procedure, henceforth termed $T$-boosting, involves no further approximations. It exploits the fact that we calculate fields $\Delta G(x)$ and $\Gamma(x)$ at the same temperature for which we eventually want to calculate the rate. We wish to stress that this virtue represents a crucial difference to temperature accelerated MD[33]. In the latter method the free energy $\Delta G(x)$ is first calculated at a high temperature and subsequently rescaled to a desired low temperature, whereupon $\Delta G(x)$ in general does change. $T$-boosting avoids this, because by using dcTMD we calculate $\Delta G(x)$ right away at the desired temperature. We note in passing that a Langevin simulation run at $T_2$ using fields obtained at $T_1$ in general does not reflect the coarse-grained dynamics of an MD simulation run at $T_2$, but can only be used to recover $k_1$ from $k_2$.

In practice, we perform $T$-boosting calculations at several temperatures $T_2$ in increments of 25 K to 50 K and choose the smallest $T_2$ such that $N \gtrsim 100$ transitions occur. In the Supporting Methods we derive an analytic expression of the extrapolation error as a function of boosting temperatures and achieved number of transitions, from which the necessary length of the individual Langevin simulations can be estimated, in order to achieve a desired extrapolation error. One-dimensional Langevin simulations require little computational effort (1 ms of simulation time at a 5 fs time step take ~6 h of wall-clock time on a single CPU) and are trivial to parallelize in the form of independent short runs. Hence the extrapolation error due to boosting can easily be pushed below 10% and is thus negligible in comparison to systematic errors coming from the dcTMD field estimates. As shown in Supplementary Table 1, a further increase in efficiency can be achieved if the considered dynamics is overdamped, which

is the case for both protein–ligand systems. Since overdamped dynamics neglects the inertia term $m\ddot{x}$ and therefore does not depend on the mass $m$, we may artificially enhance the mass in the Langevin simulations. For the protein–ligand systems, this allows us to increase the integration time step from 1 to 10 fs, i.e., a speed-up of an order of magnitude.

**Ion dissociation of NaCl in water.** To illustrate the above developed theoretical concepts and test the validity of the underlying approximations, we first consider sodium chloride in water as a simple yet nontrivial model system. For this system, detailed dcTMD as well as long unbiased MD simulations are available[22], making it a suitable benchmark system for our approach. Fig. 1a shows the free energy profiles $\Delta G(x)$ along the interionic distance $x$, whose first maximum at $x \approx 0.4$ nm corresponds to the binding-unbinding transition of the two ions. The second smaller maximum at $x \approx 0.6$ nm reflects the transition from a common to two separate hydration shells[34]. We find that results for $\Delta G(x)$ obtained from a 1 μs long unbiased MD trajectory and from dcTMD simulations ($1000 \times 1$ ns runs with $v_c = 1$ m/s) match perfectly. Since the average work $\langle W(x) \rangle$ of the nonequilibrium simulations is seen to significantly overestimate the free energy at large distances, the dissipation correction $W_{diss}$ in Eq. (2) is obviously of importance. Fig. 1b shows the underlying friction profile $\Gamma(x)$ obtained from dcTMD, which in part deviates from the lineshape of the free energy. While we also find a maximum at $x \approx 0.4$ nm, the behavior of $\Gamma(x)$ is remarkably different for larger distances $0.5 \lesssim x \lesssim 0.7$ nm, where a region of elevated friction can be found before dropping to lower values. Interestingly, these features of $\Gamma(x)$ match well the changes of the average number of water molecules bridging both ions[34]. This indicates that the increased friction in Eq. (3) is mainly caused by force fluctuations associated with the build-up of a hydration shell[22]. For $x \gtrsim 0.8$ nm, the friction is constant within our signal-to-noise resolution. The dynamics of ion dissociation and association can be described by their mean waiting times and corresponding rates shown in Fig. 2a and Table 1. For the chosen force field, ion concentration and resulting effective simulation box size, the unbiased MD simulation at 293 K yields mean dissociation and association times of $\tau_D = 1/k_D = 120$ ps and $\tau_A = 1/(k_A C) = 850$ ps, respectively, where $C$ denotes a reference concentration (see the Supplementary Methods for details). Using fields $\Delta G(x)$ and $\Gamma(x)$ obtained from TMD, the numerical integration of Langevin Eq. (1) for 1 μs results in $\tau_D = 420$ ps and $\tau_A = 3040$ ps. While the dissociation constants $K_D = k_D/k_A = 1.5$ M from Langevin and MD simulations match perfectly, we find that the Langevin predictions overestimate the correct rates by a factor of ~3.4.

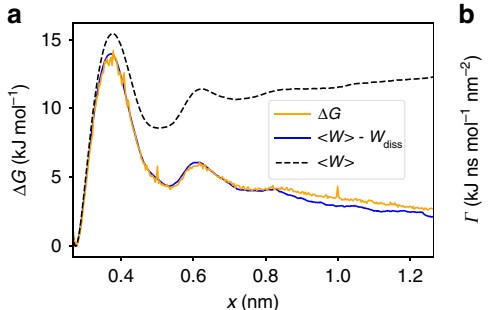
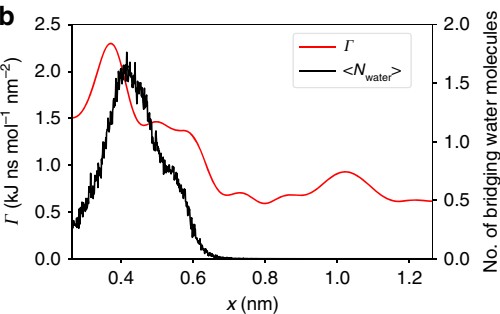

**Fig. 1 Dissociation of NaCl in water. a** Free energy profiles $\Delta G(x)$ along the interionic distance $x$, obtained from a 1 μs long unbiased MD trajectory at 293 K (orange line) and $1000 \times 1$ ns TMD runs (blue line). Error bars are given in Supplementary Fig. 2. Also shown is the average work $\langle W(x) \rangle$ calculated from the TMD simulations (dashed black line). **b** Friction profile $\Gamma(x)$ (red) obtained from dcTMD after Gaussian smoothing together with the average number of water molecules (black), that connect the $Na^+$ and $Cl^-$ ions in a common hydration shell[34].

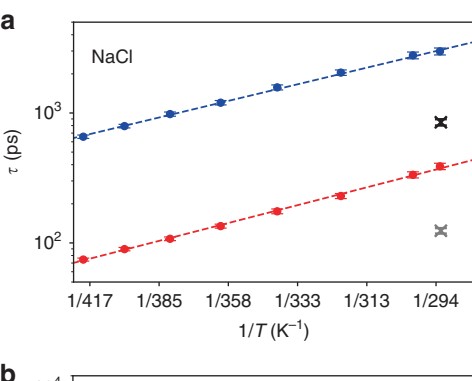

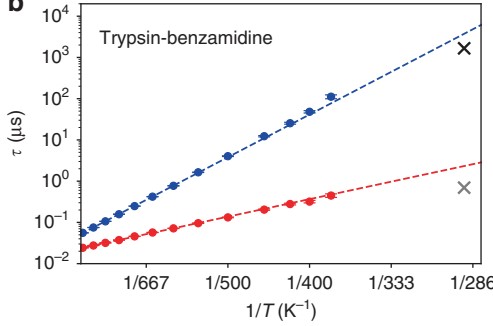

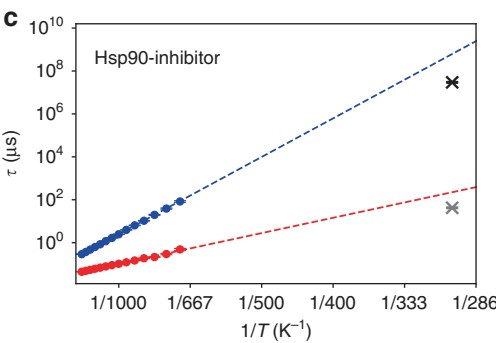

**Fig. 2 Prediction of binding and unbinding times.** Mean binding (red) and unbinding (blue) times, drawn as a function of the inverse temperature, obtained from $T$-boosted Langevin simulations of **a** solvated NaCl, **b** the trypsin-benzamidine complex, and **c** the Hsp90-inhibitor complex. Dashed lines represent fits ($R^2 = 0.90-0.99$) to Eq. (4), crosses (binding in grey, unbinding in black) indicate reference results from **a** unbiased MD simulation[22] and **b**, **c** experiment[39,48].

| Table 1 Predicted binding and unbinding kinetics. | | |
|---|---|---|
| **NaCl** | **LE** | **MD**[22] |
| $k_A$ ($10^9$ s$^{-1}$ M$^{-1}$) | 1.6 ± 0.1 | 5.5 ± 0.3 |
| $k_D$ ($10^9$ s$^{-1}$) | 2.4 ± 0.1 | 8.1 ± 0.4 |
| $K_D$ (M) | 1.5 ± 0.2 | 1.5 ± 0.2 |
| **Trypsin** | **LE** | **Experiment**[39] |
| $k_{on}$ ($10^6$ s$^{-1}$ M$^{-1}$) | 8.7 ± 0.5 | 29.0 |
| $k_{off}$ ($10^2$ s$^{-1}$) | 2.7 ± 0.4 | 6.0 |
| $K_D$ ($10^{-5}$ M) | 3.1 ± 0.6 | 2.1 |
| **Hsp90** | **LE** | **Experiment**[48] |
| $k_{on}$ ($10^4$ s$^{-1}$ M$^{-1}$) | 9.0 ± 0.8 | 48.0 ± 2.0 |
| $k_{off}$ ($10^{-3}$ s$^{-1}$) | 1.6 ± 0.2 | 34.0 ± 2.0 |
| $K_D$ ($10^{-8}$ M) | 1.8 ± 0.3 | 7.1 ± 0.5 |

Rates resulting from fits in Fig. 2 (using units of molarity M, i.e., mol/l) with fit errors as indicated[59], and reference values from unbiased MD simulations[22] and experiment[39,48], respectively. Dissociation constants were calculated from rate constants.

The latter may be caused by various issues. For one, to be of practical use, the Langevin model was deliberately kept quite simple. For example, it does not include an explicit solvent coordinate[34,35], but accounts for the complex dynamics of the solvent merely through the friction field $\Gamma(x)$. Moreover, we note that the calculation of $\Gamma(x)$ via Eq. (3) uses constraints, which have the effect of increasing the effective friction[36]. This finding is supported by calculations using the data-driven Langevin approach[37,38], which estimates friction coefficients based on unbiased MD simulations that are consistantly smaller than the ones obtained from dcTMD (Supplementary Fig. 5). Considering the simplicity of the Langevin model and the approximate calculation of the friction coefficient by dcTMD, overall we are content with a factor ~3 deviation of the predicted kinetics.

To illustrate the validity of the $T$-boosting approach suggested above, we performed a series of Langevin simulations for eight temperatures ranging from 290 to 420 K and plotted the resulting dissociation and association times as a function of the inverse temperature (Fig. 2a and Table 1). Checking the consistency of our approach, a fit to Eq. (4) yields transition state free energies $\Delta G^{\neq}$ of 13 and 12 kJ/mol for ion dissociation and association, respectively, which agree well with barrier heights of the free energy profile in Fig. 1a. Moreover, dissociation and association times obtained from the extrapolated $T$-boosted Langevin simulations ($\tau_D = 370$ ps, $\tau_A = 3050$ ps) agree excellently with the directly calculated values. This indicates that high-temperature Langevin simulations can indeed be extrapolated to obtain low-temperature transition rates.

**Trypsin-benzamidine.** Let us now consider the prediction of free energies, friction profiles and kinetics in protein–ligand systems. The first system we focus on is the inhibitor benzamidine bound to trypsin[39–41], which represents a well-established model problem to test enhanced sampling techniques[21,31,42–45]. The slowest dynamics in this system is found in the unbinding process, which occurs on a scale of milliseconds[39]. To capture the kinetics of the unbinding process, so far Markov state models[42,43], metadynamics[31], Brownian dynamics[44] and adaptive enhanced sampling methods[21,45] have been employed.

Here we combined dcTMD simulations and a subsequent nonequilibrium principal component analysis[30] to identify the dominant dissociation pathways of ligands during unbinding from their host proteins (see Supplementary Methods). Fig. 3 shows TMD snapshots of the structural evolution along this pathway, its free energy profile $\Delta G(x)$, and the associated friction $\Gamma(x)$. Starting from the bound state ($x_1 = 0$ nm), $\Delta G(x)$ exhibits a single maximum at $x_2 \approx 0.46$ nm, before it reaches the dissociated state for $x \gtrsim x_4 = 0.75$ nm. In line with the findings of Tiwary et al.[31], the maximum of $\Delta G(x)$ reflects the rupture of the Asp189-benzamidine salt bridge, which represents the most important contact of the bound ligand. Following right after, the friction profile $\Gamma(x)$ reaches its maximum at $x_3 \approx 0.54$ nm, where the charged side chain of benzamidine becomes hydrated with water molecules. Similarly to NaCl, the friction peak coincides with the increase in the average number of hydrogen bonds between benzamidine and bulk water. The peak in friction is slightly shifted to higher $x$, because the ligand acts as a plug for the binding site, and first needs to be (at least partially) removed in order to allow water flowing in. As for the dissociation of NaCl in water, enhanced friction during unbinding appears to be directly linked to a rearrangement of the protein–ligand hydration shell, which is in agreement with recent results from neutron crystallography[41].

To calculate rates $k_{on}$ and $k_{off}$ describing the binding and unbinding of benzamidine from trypsin, we performed 10 ms long

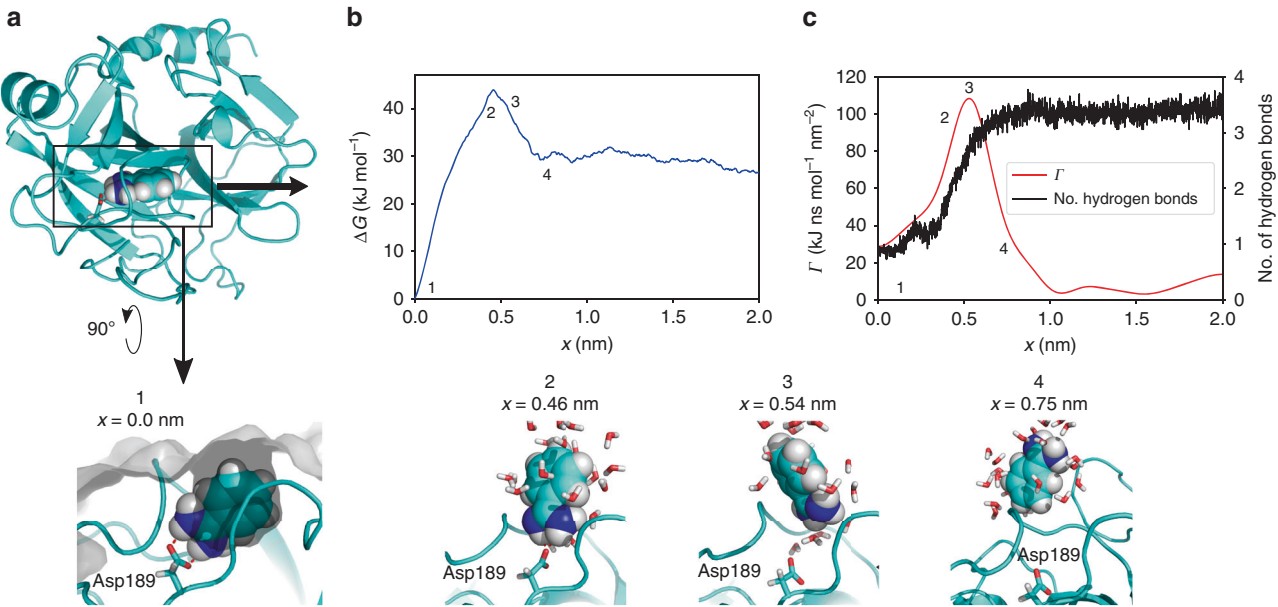

**Fig. 3 Unbinding of benzamidine from trypsin. a** TMD snapshots of the structural evolution in trypsin along the dominant dissociation pathway, showing protein surface in gray, benzamidine as van der Waals spheres, Asp189 and water molecules as sticks. Benzamidine is bound to the protein in a cleft of the protein surface via a bidental salt bridge to Asp189. dcTMD calculations of **b** free energy $\Delta G(x)$, and **c** (Gaussian smoothed) friction $\Gamma(x)$ together with the mean number of hydrogen bonds between benzamidine and water. Highlighted are the bound state 1, transition state 2, the state with maximal friction 3 and the unbound state 4. Error bars of free energy and friction estimates are given in Supplementary Fig. 2.

Langevin simulations along the dominant pathways at thirteen temperatures ranging from 380–900 K. As shown in Fig. 2b and Table 1, the resulting rates are well fitted ($R^2 \geq 0.90$) by the $T$-boosting expression in Eq. (4). Representing the resulting number of transitions as a function of the inverse temperature, we find that at 380 K only ~9 events happen during a millisecond. That is, to obtain statistically converged rates at 290 K would require Langevin simulations at 290 K on a timescale of seconds. Using temperature boosting with Eq. (4), on the other hand, our high-temperature millisecond Langevin simulations readily yield converged transition rates at 290 K (see Fig. 2b and Table 1), that is, $k_{\text{on}} = 8.7 \times 10^6\,\text{s}^{-1}\,\text{M}^{-1}$ and $k_{\text{off}} = 2.7 \times 10^2\,\text{s}^{-1}$, which underestimate the experimental values[39] $k_{\text{on}} = 2.9 \times 10^7\,\text{s}^{-1}\,\text{M}^{-1}$ and $k_{\text{off}} = 6.0 \times 10^2\,\text{s}^{-1}$ by a factor of 2–3. Similarly, the calculated $K_D$ overestimates the experimental result[39] of $K_D = 2.1 \times 10^{-5}$ M by a factor of ~1.5. As indicated by a recent review[3] comparing numerous computational methods to calculate (un)binding rates of trypsin-benzamidine, our approach compares quite favorably regarding accuracy and computational effort.

As the extrapolation error due to $T$-boosting is negligible (see Supplementary Methods), the observed error is mainly caused by the approximate calculation of free energy and friction fields by dcTMD. In the case of NaCl, we have shown that reliable estimates of the fields (with errors $\lesssim 1\,k_BT$) require an ensemble of at least 500 simulations (see ref. [22] and Supplementary Fig. 2), although the means of $\Delta G$ and $\Gamma$ appear to converge already for ~100 trajectories. In a similar vein, by performing a Jackknife "leave-one-out" analysis[46], for trypsin-benzamidine we obtain an error of ~$2\,k_BT$ for 150 trajectories (Supplementary Fig. 2). Interestingly, the error of the main free energy barrier is typically comparatively small, because the friction and thus variance of $W$ increase directly after the barrier. As a consequence, the sampling error of $k_{\text{off}}$ is small compared to that of $k_{\text{on}}$ and the binding free energy. We note that if the experimental binding affinity $K_D$ is known, it can be used as a further constraint on the error of the free energy and friction fields.

**Hsp90-inhibitor**. The second investigated protein complex is the N-terminal domain of heat shock protein 90 (Hsp90) bound to a resorcinol scaffold-based inhibitor (**1j** in ref. [47]). This protein has recently been established as a test system for investigating the molecular effects influencing binding kinetics[47–50], and the selected inhibitor unbinds on a scale of half a minute. From the overall appearance of free energy and friction profiles (Fig. 4), we observe clear similarities to the case of trypsin-benzamidine. That is, the main transition barrier is also found at $x_2 \approx 0.5$ nm, which stems from the ligand pushing between two helices at this point in order to escape the binding site. Moreover, the friction peaks at $x_2 \approx 0.5$ nm, as well, but with an additional shoulder at $x_3 \approx 0.8$ nm, which again coincides with changes of the ligand's hydration shell. The unbound state is reached after $x \gtrsim 1.0$ nm. We note that the ligand is again bound to the protein via a hydrogen bond to an aspartate (Asp93) and at a position that is open to the bulk water.

To calculate rates $k_{\text{on}}$ and $k_{\text{off}}$, we again performed 5 ms long Langevin simulations along the dissociation pathway at fourteen different temperatures ranging from 700–1350 K. Rate prediction (see Fig. 2c and Table 1) yields $k_{\text{on}} = 9.0 \times 10^4\,\text{s}^{-1}\,\text{M}^{-1}$ and $k_{\text{off}} = 1.6 \times 10^{-3}\,\text{s}^{-1}$, and underestimates the experimental[48] values $k_{\text{on}} = 4.8 \pm 0.2 \times 10^5\,\text{s}^{-1}\,\text{M}^{-1}$ and $k_{\text{off}} = 3.4 \pm 0.2 \times 10^{-2}\,\text{s}^{-1}$ by a factor of 5–20. The resulting value for $K_D = 1.8 \times 10^{-8}$ M underestimates the experimental value[48] $7.1 \times 10^{-8}$ M by a factor of ~4. Considering that we attempt to predict unbinding times on a time scale of half a minute from sub-$\mu$s MD simulations, and that a factor 20 corresponds to a free energy difference of about $3\,k_BT$ (i.e., 15 % of the barrier height in Hsp90), we find this agreement remarkable for a first principles approach which implies many uncertainties of the physical model[51]. We attribute the larger deviation in comparison to trypsin to issues with the sampling of the correct unbinding pathways: especially unbinding rates in the range of minutes to hours fall into the same timescale as slow conformational dynamics of host proteins[48], requiring a sufficient sampling of the conformational space of the protein as a prerequisite for dcTMD pulling simulations.

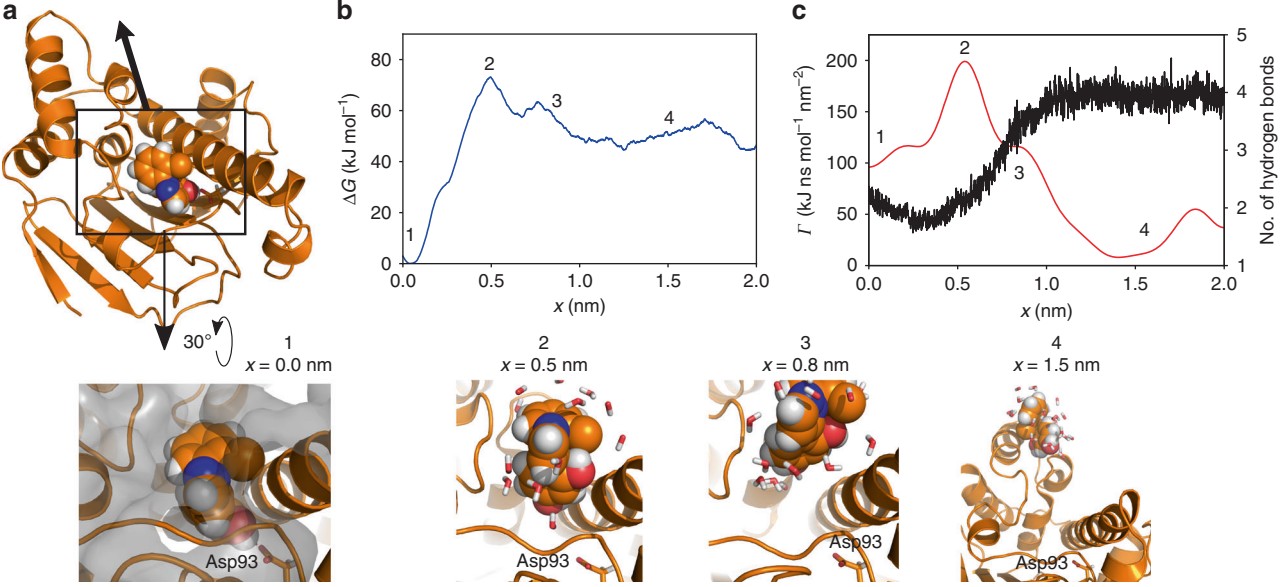

**Fig. 4 Unbinding of an inhibitor from the N-terminal domain of Hsp90. a** Structural evolution along the dissociation pathway in Hsp90, showing protein surface in gray, inhibitor as van der Waals spheres, Asp93 and water molecules as sticks. The inhibitor is bound to the protein in a cleft of the protein surface via a hydrogen bond to Asp93. dcTMD calculations of **b** free energy $\Delta G(x)$ and, **c** (Gaussian-smoothed) friction $\Gamma(x)$ together with the mean number of hydrogen bonds between inhibitor and water. Highlighted are the bound state 1, transition state and state with maximal friction 2, an additional state with increased friction 3 and the unbound state 4. Error bars of free energy and friction estimates are given in Supplementary Fig. 2. Fluctuations of $\Gamma(x)$ for $x \gtrsim 1$ nm are due to noise. Color code as in Fig. 3.

## Discussion

Using free energy and friction profiles obtained from dcTMD, we have shown that $T$-boosted Langevin simulations yield binding and unbinding rates which are well comparable to results from atomistic equilibrium MD and experiments. That is, rates are underestimated by an order of magnitude or less which, in comparison to other methods that have been applied to the trypsin-benzamidne and Hsp90 complexes (see refs. [3,52] for recent reviews), is within the top accuracy currently achievable. At the same time, the few other methods that aim at predicting absolute rates (such as Markov state models[42,43] and infrequent metadynamics[31,53]) require substantial more MD simulation time, while dcTMD only requires sub-$\mu$s MD runs, that is, at least an order of magnitude less computational time. As the extrapolation error due to $T$-boosting is negligible, the error is mainly caused by the approximate calculation of free energy and friction fields by dcTMD. We have shown that friction profiles, which correspond to the dynamical aspect of ligand binding and unbinding, may yield additional insight into molecular mechanisms of unbinding processes, which are not reflected in the free energies. Although the three investigated molecular systems differ significantly, in all cases friction was found to be governed by the dynamics of solvation shells.

## Methods

**MD simulations**. All simulations employed Gromacs v2018 (ref. [54]) in a CPU/GPU hybrid implementation, using the Amber99SB* force field[55,56] and the TIP3P water model[57]. For each system, $10^2$–$10^3$ dcTMD calculations[22] at pulling velocity $v_c = 1$ m/s were performed to calculate free energy $\Delta G(x)$ and friction $\Gamma(x)$. For the NaCl-water system, dcTMD as well as unbiased MD simulations were taken from ref. [22]. Trypsin-benzamidin complex simulations are based on the 1.7 Å X-ray crystal structure with PDB ID 3PTB[40]. Simulation systems of the Hsp90-inhibitor complex were taken from ref. [47]. Detailed information on system preparation, ligand parameterization, MD simulations and pathway separation can be found in the Supplementary Methods.

**Langevin simulations**. Langevin simulations employed the integration scheme by Bussi and Parrinello[58]. Details on the performance of this method with respect to the employed integration time step and system mass can be found in the Supplementary Methods.

## Data availability

Simulation data on NaCl, Trypsin-benzamidine, and Hsp90-inhibitor is available from the authors upon request.

## Code availability

Python scripts for dcTMD calculations, the fastpca program package for nonequilibrium principal component analysis, the data-driven Langevin package, the Langevin simulation code, and Jupyter notebooks for $T$-boosting analysis and sampling error estimation in Langevin simulations are available at our website www.moldyn.uni-freiburg.de.

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

## Acknowledgements

We thank Peter Hamm and Matthias Post for numerous instructive and helpful discussions. The authors acknowledge support by the Deutsche Forschungsgemeinschaft (Sto 247/11), by the bwUniCluster computing initiative, the High Performance and Cloud Computing Group at the Zentrum für Datenverarbeitung of the University of Tübingen, the state of Baden-Württemberg through bwHPC and the Deutsche Forschungsgemeinschaft through grant No. INST 37/935-1 FUGG, and the Freiburg Institute for Advanced Studies (FRIAS) of the Albert-Ludwigs-University Freiburg. The article processing charge was partially funded by the Albert-Ludwigs-University Freiburg in the funding programme Open Access Publishing.

## Author contributions

S.W. and G.S. designed and supervised research. S.W. performed TMD and Langevin simulations and nonequilibrium path separation of Trypsin trajectories. B.L. performed dLE analysis and implemented Langevin simulations. S.B. performed the nonequilibrium path separation of Hsp90 trajectories. All authors wrote the paper.

## Competing Interests

The authors declare no competing interests.
