## [Peer Review File · Nature Communications]

Reviewers' comments:

Reviewer #1 (Remarks to the Author):

The work by Wolf et al is an interesting and possibly very practical approach to obtain multi second ligand dissociation dynamics from atomistic simulations. I do have many concerns about several crucial details in this paper. It is possible that after these details are addressed, the paper is suitable for publication in Nature Commun. However in its current state, it lacks somewhat in both rigor and novelty. Again - these concerns could be overcome in a revised version and I request the authors to address them. The manuscript has the potential to be a game changer if these could be dealt with.

My concerns are:

1. I will start with the statement in the first column of second page "As discussed in Ref. 20, the derivation of Langevin...equilibrium at all times". This is not a trivial requirement and I believe it will depend heavily on the choice of x in Eq 1, as good x (close to "true" reaction coordinate") could arguably allow high pulling speeds, as all other orthogonal directions would per construction be fast equilibrators. For a bad choice of x , the other directions could be extremely slow at equilibrating and any pulling speed which maintains "equilibrium at all times" would be impractical to achieve. I mention alanine dipeptide in vacuum as an example where $x = \phi$, the "good" dihedral angle, this statement would be fine, but if $x = \psi$ I believe it will become impractical. The authors need to address this carefully, especially as they have not put any effort into describing x in this work for the complex systems. It would have been nice to see detailed quantitative analysis and heuristics on this in the supplement, before any one applies such a nice and simple method in complicated drug discovery efforts for example. How would they pick x , and how would they ascertain a slow enough pulling speed?

2. Related, the Jarzynski fluctuation relation is notorious for convergence problems, see for example Bhattacharjee, Kirkpatrick, Sengers PRE <https://journals.aps.org/pre/abstract/10.1103/PhysRevE.97.042109> and Vaikuntanathan, Jarzynski, PRL <https://journals.aps.org/prl/abstract/10.1103/PhysRevLett.100.190601> The systems considered here are equally if not more complex, why should then one trust convergence of Jarzynski-based estimates of the free energy and especially so the friction? Maybe this was addressed in their prior work as to how they managed to break past this curse of poor convergence. If so, that is important enough to be again described here at least briefly.

3. Eq. 4 is critical to this work as the simulations themselves are not performed at temperature of interest, but at higher temperatures and then extrapolated to lower temperature. This equation is reminiscent of Voter's temperature accelerated dynamics. It would be apt to compare with that method. There Voter spends a lot of effort in determining different paths with different ΔG (barrier) and then extrapolating to low temperatures and selecting preferred path. Here it seems the authors assume a constant ΔG (I am omitting barrier for brevity), which the Langevin simulation conforms to, but that does not confirm the validity of constant ΔG . Why should constant ΔG be true? What if there are many pathways with different ΔG s? Perhaps this method is apt for koff along one pathway, which might or might not be the true pathway. Perhaps it is more generic and the constant ΔG assumption is not as weak as I am making it to be. In either case a decent discussion is warranted here.

In summary, this manuscript has unique potential to be important and useful, and I hope the authors address my concerns.

- Pratyush Tiwary, University of Maryland

Reviewer #2 (Remarks to the Author):

Wolf and colleagues present a combined approach to compute rates and rate-limiting steps in protein-ligand binding. While computing relative and absolute ligand binding free energies is increasingly feasible, thanks to recent progress in force-fields and a number of alchemical and path-based enhanced sampling algorithms, computing accurate binding kinetics is still challenging. The approach proposed by Wolf et al., namely a combination of dissipation-corrected targeted molecular dynamics (dcTMD) simulations and temperature-boosted Langevin simulations, is able to compute rates for simple ligand binding events that are within a factor of 2 to 10 from the experiments. It also predicts that binding and unbinding dynamics are mediated by changes of the surrounding hydration shell, in line with previous reports. The methods will be surely of interest to a specialized audience. However, it is not clear whether it is sufficiently novel for Nature Communications. A number of methods (including, but not limited to those cited in the manuscript) already reproduced more or less correctly the rates associated with the simple systems investigated here. The main component of the method proposed (dcTMD) has been already published elsewhere and, perhaps more concerning, it shows its limitation already on Hsp90, which has a somewhat more complex binding mechanism than the others investigated here. Thus, in its current form, the manuscript is probably more suitable for a specialized journal.

Reviewer #3 (Remarks to the Author):

In this manuscript, Wolf et al. present a computational approach to study dissociation dynamics based on preliminary targeted molecular dynamics followed by temperature-accelerated Langevin simulations. The topic is certainly of high interest to a wide community, in particular to those researchers studying small-molecules binding to protein targets. However, there are currently a number of different methods to address this problem and this manuscript does not present strong evidences that the proposed approach is more accurate, precise, or computationally convenient than the state-of-the-art in the field. Therefore, I cannot recommend the manuscript for publication in Nature Communications in the current form. I would be open to revise my position provided that additional work aimed at demonstrating advantages and disadvantages of the proposed approach is carried out.

Specific issues:

1) The sentence "The accuracy of our method is comparable to results of the best available enhanced sampling methods, although it only requires sub- μ s MD runs and relatively inexpensive Langevin simulations" is the only mention of the relative performances of this method compared to the state-of-the-art. I found this highly unsatisfying and I invite the authors to select a few of the alternative computational methods to calculate association/dissociation dynamics, carry out the calculations, and compare their results in terms of accuracy, precision, and computational cost. I invite the authors to carry out these comparisons in all the 3 test cases reported: dissociation of NaCl in water, Trypsin-benzamidine, and Hsp90-inhibitor. Among the methods available, I am particularly curious to see the performances against the metadynamics-based approach proposed by Tiwary and Parrinello, in which information about the dynamics is directly recovered from the biased simulations. In case the method proposed by the authors provided a leap-forward compared to the existing approaches, I would support its publication in Nature Communications.

2) I invite the authors to report error bars in the main manuscript, both in all the free-energy profiles of Figures 1, 3 and 4 and in all free-energy differences reported in the text. For example, I find the sentence on page 5 "The predicted free energy in the unbound state of ~

45 kJ/mol compares well to the experimental value of 40.7 ± 0.2 kJ/mol" quite inappropriate without an estimate of the error in the prediction.

3) The free-energy profile as a function of the distance x reported in Figure 1A seems to show the correct behavior at long distances, i.e. it goes to minus infinity as $-\log(x^2)$.

On the contrary, the profiles in Figures 3B and 4B are quite flat at long distances. Could the authors comment on this point?

Also, could the authors explain how they calculated the free-energy differences unbound/bound in the case of trypsin (27 kJoule/mol) and Hsp90 (45 kJoule/mol).

By looking at the corresponding free-energy profiles as a function of the distance collective variable, it appears that this difference is just the value of the free-energy in an arbitrary point of the unbound state(s), which would be incorrect - missing concentration correction and "entropy" term $-\log(x^2)$. This would also be problematic in the case of Hsp90 inhibitor, in which there are large oscillations in the free-energy profile beyond 1.0 nm.

Resubmission of

”Multisecond ligand dissociation dynamics from atomistic simulations”

by Wolf et al.

Comments of Reviewer 1

1. I will start with the statement in the first column of second page As discussed in Ref. 20, the derivation of Langevinequilibrium at all times. This is not a trivial requirement and I believe it will be depend heavily on the choice of x in Eq 1, as good x (close to true reaction coordinate) could arguably allow high pulling speeds, as all other orthogonal directions would per construction be fast equilibrators. For a bad choice of x , the other directions could be extremely slow at equilibrating and any pulling speed which maintains equilibrium at all times would be impractical to achieve. I mention alanine dipeptide in vacuum as an example where is $x=\phi$, the good dihedral angle, this statement would be fine, but if $x=\psi$ I believe it will become impractical. The authors need to address this carefully, especially as they have not put any effort into describing x in this work for the complex systems. It would have been nice to see detailed quantitative analysis and heuristics on this in the supplement, before any one applies such a nice and simple method in complicated drug discovery efforts for example. How would they pick x , and how would they ascertain a slow enough pulling speed?

Author Reply: We do agree with the reviewer that the choice of x and the pulling velocity is not trivial and needs to be carried out with caution. In fact, the choice of the reaction coordinate represents a general problem that persist for most types of biased simulation techniques. In our case, there are several aspects to be discussed:

For one, considering ligand-protein unbinding, we generally use the distance between the center of masses of C(α) atoms of the central beta-sheet of the protein on one hand and ligand heavy atoms on the other hand as simple biasing coordinate. The approach has been found to result in a promising and simple biasing coordinate (Wolf et al, Ref. [49]). This is now described in detail in the SI for all cases.

Secondly and more importantly, when using a 1D pulling coordinate, we typically find that the ligands take several distinct pathways out of the protein binding sites due to diffusion perpendicular to the biasing coordinate. Here, we distinguish the different pathways via a nonequilibrium PCA (Post et al, Ref. [30]), perform a clustering of trajectories according to these pathways, and consider in the subsequent Langevin simulation typically only the most important pathway with the lowest barrier energy. In case of similar barrier heights, we use the pathway that is taken by the majority of biased simulation trajectories. A similar strategy is also followed by Tiwary et al. (Ref. [31]) who used preliminary funnel metadynamics runs, followed by a protein-ligand contact map analysis to construct suitable path collective variables. The separation of pathways is also crucial to warrant that the work is Gaussian distributed, which is required to safely apply a second-order cumulant expansion of Jarzynski’s identity (see answer to point 2 below).

Apart from explaining this important point in the last paragraph of section Theory:dcTMD, we now - added to this paragraph the sentence: “This approach bears similarities with the work of Tiwary et al. for the construction of path collective variables³¹.”

- significantly extended the corresponding section of the Supplementary Information to include details on pathway separation,

- and included the new Supplementary Figure 3 that illustrates the path separation.

Thirdly, the choice of the pulling velocity is also closely related to the path separation. That is, we observe that a velocity of 1 m/s appears to result in an optimal number of pathways: at higher velocities, trajectories do not cluster into pathways but smear out, while at slower velocities, recrossing between pathways appears due to artificial stationarity. This is now discussed in the Supplementary Information, which also shows the new Supplementary Figure 4.

In the main text, we added at the end of last paragraph of section Theory:dcTMD, “Details on the convergence of the free energy estimate [Eq. (2)], the path separation and the choice of the pulling velocity are given in the Supplementary Methods and Supplementary Figs. 1 - 4).”

2. Related, the Jarzynski fluctuation relation is notorious for convergence problems, see for example Bhattacharjee, Kirkpatrick, Sengers PRE <https://journals.aps.org/pre/abstract/10.1103/PhysRevE.97.042109> and Vaikuntanathan, Jarzynski, PRL <https://journals.aps.org/prl/abstract/10.1103/PhysRevLett.100.190601>. The systems considered here are equally if not more complex, why should then one trust convergence of Jarzynski-based estimates of the free energy and especially so the friction? Maybe this was addressed in their prior work as to how they managed to break past this curse of poor convergence. If so, that is important enough to be again described here at least briefly.

Author Reply: The well-known convergence problems associated with Jarzynski’s identity are a consequence of the fact that the free energy estimates strongly depend on the sampling of low-work trajectories, and exhibit an erratic convergence due to saturation effects from events in the central distribution and the high-work tail. We circumvent this problem by assuming a normal distribution of the work, which allows us to apply a second-order cumulant expansion of Jarzynski’s identity. This gives the dissipated work in terms of the variance of the nonequilibrium work, which is much easier to compute and converge.

To clarify this point,

- we added to the second paragraph of section Theory:dcTMD: “...we invoke Jarzynski’s identity... To circumvent convergence problems associated with the above exponential average²⁷, we perform a second-order cumulant expansion which gives ...”

- we stress this point in the last paragraph of section Theory:dcTMD: “Secondly, to warrant rapid convergence, we have invoked a cumulant expansion to derive the friction coefficient ...”

- we added Supplementary Figure 1 together with additional explanatory text to the SI, which demonstrates the improvement of the convergence of the cumulant approximation compared to the direct evaluation of Jarzynski’s identity.

3. Eq. 4 is critical to this work as the simulations themselves are not performed at temperature of interest, but at higher temperatures and then extrapolated to lower temperature. This equation is reminiscent of Voter's temperature accelerated dynamics. It would be apt to compare with that method. There Voter spends a lot of effort in determining different paths with different ΔG (barrier) and then extrapolating to low temperatures and selecting preferred path. Here it seems the authors assume a constant ΔG (I am omitting barrier for brevity), which the Langevin simulation conforms to, but that does not confirm the validity of constant ΔG . Why should constant ΔG be true? What if there are many pathways with different ΔG s? Perhaps this method is apt for koff along one pathway, which might or might not be the true pathway. Perhaps it is more generic and the constant ΔG assumption is not as weak as I am making it to be. In either case a decent discussion is warranted here.

Author Reply: There is a crucial difference between Voter's method and what we here called T-boosting, which is admittedly somewhat tricky to understand. In temperature accelerated MD by Voter, typically the free energy $\Delta G(x)$ is first calculated at some high temperature. When rescaling to the desired temperature such as 300 K, the free energy $\Delta G(x)$ in general does change, as the underlying partition function is a function of T . In our approach we don't do that, because by using dcTMD we calculate $\Delta G(x)$ right away at the desired temperature. Here is another attempt to explain our train of thought:

(1) Let us consider an unbiased Langevin model (Eq. (1) with $f_c = 0$) of some process of interest. The model is defined by the free energy $\Delta G(x)$ and friction $\Gamma(x)$, which we assume to be given. Running simulations at various temperatures $T_1, T_2 \dots$ will result in corresponding transition rates $k_1, k_2 \dots$ of the considered process. Employing Kramers' famous expression, we can easily derive Eq. (4) of the paper, which relates the transition rate k_1 for temperature T_1 to the transition rate k_2 for temperature T_2 . That is, the prefactor of the rates cancels out, and the rates are simply related by the corresponding Boltzmann factors.

(2) In T-boosting, we first use dcTMD to obtain free energy $\Delta G(x)$ and friction $\Gamma(x)$, which are the input needed to construct a Langevin model (Eq. (1) with $f_c = 0$) of the considered process. This Langevin equation describes the dynamics at the temperature the dcTMD simulations were run to calculate $\Delta G(x)$ and $\Gamma(x)$, that is, at the desired temperature T_1 (usually 300 K). Hence just as done in (1), we may propagate this Langevin equation at higher temperatures T_2 and apply Eq. (4) to calculate the transition rate k_1 for temperature T_1 from the transition rate k_2 for temperature T_2 . Since $\Delta G(x)$ and $\Gamma(x)$ were calculated at the same temperature for which we eventually want to calculate the rate, T-boosting does not involve an additional approximation since we do not claim to predict the right dynamics at T_2 .

To clarify this issue, we rewrote section Theory:T-boosting and in particular added to the fourth paragraph:

“[T-boosting] exploits the fact that we calculate fields $\Delta G(x)$ and $\Gamma(x)$ at the same temperature for which we eventually want to calculate the rate. We wish to stress that this virtue represents also a crucial difference to temperature accelerated MD³⁴. Here the free energy $\Delta G(x)$ is first calculated at high temperature and subsequently rescaled to a desired low temperature, whereupon $\Delta G(x)$ in

general does change. T -boosting avoids this, because by using dcTMD we calculate $\Delta G(x)$ right away at the desired temperature.”

Concerning the presence of multiple possible pathways, we agree with the reviewer. This problem is taken care of by us via our pathway separation method (see our reply to 2.). We further note that the search of the right unbinding pathway is related to the search of the right collective variable describing unbinding, and we are currently carrying out research to search for optimal (or, at least, sufficiently good) unbinding coordinates and methods for determining the most probable unbinding path.

Comments of Reviewer 2

Wolf and colleagues present a combined approach to compute rates and rate-limiting steps in protein-ligand binding. While computing relative and absolute ligand binding free energies is increasingly feasible, thanks to recent progress in force-fields and a number of alchemical and path-based enhanced sampling algorithms, computing accurate binding kinetics is still challenging. The approach proposed by Wolf et al., namely a combination of dissipation-corrected targeted molecular dynamics (dcTMD) simulations and temperature-boosted Langevin simulations, is able to compute rates for simple ligand binding events that are within a factor of 2 to 10 from the experiments. It also predicts that binding and unbinding dynamics are mediated by changes of the surrounding hydration shell, in line with previous reports. The methods will be surely of interest to a specialized audience. However, it is not clear whether it is sufficiently novel for Nature Communications. A number of methods (including, but not limited to those cited in the manuscript) already reproduced more or less correctly the rates associated with the simple systems investigated here. The main component of the method proposed (dcTMD) has been already published elsewhere and, perhaps more concerning, it shows its limitation already on Hsp90, which has a somewhat more complex binding mechanism than the others investigated here. Thus, in its current form, the manuscript is probably more suitable for a specialized journal.

Author Reply: The general concern that our manuscript is “more suitable for a specialized journal” prompted us to restate the significance of our work in the Conclusions of the revised paper. We now write:

“Using free energy and friction profiles obtained from dcTMD, we have shown that T -boosted Langevin simulations yield binding and unbinding rates which are well comparable to results from atomistic equilibrium MD and experiments. That is, rates are underestimated by an order of magnitude or less which, in comparison to other methods that have been applied to the trypsin-benzamidne and Hsp90 complexes (see Refs. 3 and 53 for recent reviews), is within the top accuracy currently achievable. At the same time, the few other methods that aim at predicting absolute rates (such as Markov state models^{44,45} and infrequent metadynamics^{31,54}) require substantial more MD simulation time, while dcTMD only requires sub- μ s MD runs, that is, at least an order of magnitude less computational time. As the extrapolation error due to T -boosting is negligible, the error is mainly caused by the approximate calculation of free energy and friction fields by dcTMD. We have shown

that friction profiles, which correspond to the nonequilibrium aspect of ligand binding and unbinding, may yield additional insight into molecular mechanisms of unbinding processes, which are not reflected in the free energies. Although the three investigated molecular systems differ significantly, in all cases friction was found to be governed by the dynamics of solvation shells.”

While numerous methods for the determination of free energy profiles of unbinding events do certainly exist, we wish to stress that our method is to our knowledge the only approach that is based on a nonequilibrium statistical mechanics ansatz to calculate free energies *and* friction factors. The latter reflect dynamics of all remaining degrees of freedom in the system that the free energy is oblivious to.

Concerning the more specific concern, that the method “shows limitation already for Hsp90”, we wish to stress that an order of magnitude deviation between experimental and calculated unbinding rate for such a system is not only superior to other methods (see answer above), but is in fact remarkable for a first principles approach. To clarify this point, we write now in the last paragraph before the Conclusions: “Considering that we attempt to predict unbinding times on a time scale of half a minute from sub- μ s MD simulations, and that a factor 20 corresponds to a free energy difference of about $3 k_B T$ (i.e., 15 % of the barrier height in Hsp90), we find this agreement remarkable for a first principles approach which implies many uncertainties of the physical model.” We also added a citation of a recent preprint from the Carloni group (Ref. 52) addressing such uncertainties.

Comments of Reviewer 3

1. The sentence “The accuracy of our method is comparable to results of the best available enhanced sampling methods, although it only requires sub- μ s MD runs and relatively inexpensive Langevin simulations” is the only mention of the relative performances of this method compared to the state-of-the-art. I found this highly unsatisfying and I invite the authors to select a few of the alternative computational methods to calculate association/dissociation dynamics, carry out the calculations, and compare their results in terms of accuracy, precision, and computational cost. I invite the authors to carry out these comparisons in all the 3 test cases reported: dissociation of NaCl in water, Trypsin-benzamidine, and Hsp90-inhibitor. Among the methods available, I am particularly curious to see the performances against the metadynamics-based approach proposed by Tiwary and Parrinello, in which information about the dynamics is directly recovered from the biased simulations. In case the method proposed by the authors provided a leap-forward compared to the existing approaches, I would support its publication in Nature Communications.

Author Reply:

First off, we agree with the reviewer that we need to strengthen this point. Hence we have rewritten the Conclusions (see answer to Reviewer 2 above), which now explains in more detail the virtues of our approach compared to other methods.

Secondly, we note that the above sentence was meant to point to a recent review (Bruce et al., Ref. [3]), which compares numerous computational methods to calculate free energies and (un)binding

rates of various ligand-protein systems, including trypsin-benzamidine. For the latter, metadynamics was quoted to underestimate the unbinding rate by a factor of 70 and require an accumulated trajectory length of 5 μs , which is to be compared with an underestimation by a factor of 2 and an accumulated trajectory length of 0.4 μs for our method. To emphasize the reference to the review, we now added to the third paragraph of the trypsin-benzamidine section: “As indicated by a recent review³ comparing numerous computational methods to calculate (un)binding rates of trypsin-benzamidine, our approach compares quite favorably regarding accuracy and computational effort.”

Concerning Hsp90, our calculation interestingly represents the first computational prediction of absolute binding and unbinding rates for this well-established drug target for cancer treatment (see the recent review of Nunes-Alves et al., Ref. [53]). While a metadynamics study on this system might be nice to do in separate future works, it is a project of its own at this stage and clearly beyond the scope of this work. On the other hand, metadynamics has been used to calculate these rates for ligand-protein systems of similar complexity and unbinding rates on the order of seconds, see e.g. the recent study on a p38 MAP kinase of Parrinello and coworkers [Ref. 54]. Again we find comparable accuracy of the rates (a factor of 7 off) but a clearly higher computational effort (at least an order of magnitude) compared to our study of Hsp90. Finally, we note that NaCl exhibits dynamics in the upper picosecond- to lower nanosecond range and thus does not constitute a suitable system to be treated by metadynamics (see Ref. [19], Tiwary and Parrinello, PRL 2013, comment on alanine dipeptide as test system in the 2nd paragraph on p. 230602-4).

2. I invite the authors to report error bars in the main manuscript, both in all the free-energy profiles of Figures 1, 3 and 4 and in all free-energy differences reported in the text. For example, I find the sentence on page 5 “The predicted free energy in the unbound state of 45 kJ/mol compares well to the experimental value of 40.7 ± 0.2 kJ/mol” quite inappropriate without an estimate of the error in the prediction.

Author Reply: We certainly agree with the reviewer that a solid discussion of the various errors is essential, when a new numerical method is established. To this end, the requested error bars for all the free-energy profiles together with an error analysis of all friction profiles are given in Supplementary Figure 6. Besides in the main text, we now also refer in the captions of Figs. 1, 3 and 4 to this additional figure. We choose the SI for this purpose, because an adequate error analysis necessitates some discussion (we use a Jackknife analysis here) and also requires showing results for several ensemble sizes.

We also agree that the mentioned sentence on the comparability of experimental and theoretical values is of little significance. Hence we changed the sentence into a comparison of dissociation constants (see also the reply to point 3 below) and added computational error bars for both trypsin-benzamidine and Hsp90.

3. The free-energy profile as a function of the distance x reported in Figure 1A seems to show the correct behavior at long distances, i.e. it goes to minus infinity as $-\log(x^2)$. On the contrary, the profiles in Figures 3B and 4B are quite flat at long distances. Could the authors comment on this point?

Author Reply: We note that the depth of the potential well in the bound state of NaCl is not very low in relation to the unbound state. This is not the case for the two protein-ligand systems, since the ligands exhibit additional enthalpic contributions from van der Waals interactions and the hydrophobic effect at short protein-ligand distances. The observed decay of $-\log(x^2)$ to zero comes from an entropic effect, which only contributes 2.5 kJ/mol over 0.8 nm in the case of NaCl (judged on the decay of the free energy curve after the main energy barrier). An apparent decay on a similar scale can be observed for the two protein-ligand systems in Figs. 3B and 4B after the main barriers, as well. The absolute extent of this decay, though, is lost due to the uncertainty of our method.

Also, could the authors explain how they calculated the free-energy differences unbound/bound in the case of trypsin (27 kJoule/mol) and Hsp90 (45 kJoule/mol). By looking at the corresponding free-energy profiles as a function of the distance collective variable, it appears that this difference is just the value of the free-energy in an arbitrary point of the unbound state(s), which would be incorrect - missing concentration correction and "entropy" term $-\log(x^2)$. This would also be problematic in the case of Hsp90 inhibitor, in which there are large oscillations in the free-energy profile beyond 1.0 nm.

Author Reply: This comparison is based on the free energy difference at the minimum around $x \approx 0$ nm and the maximum around $x = 2$ nm. We agree that the direct comparison of our coordinate-dependent free energy profiles and standard free energies of binding from experiment, which we calculated as $\Delta G = -k_B T \ln(K_D/C_0)$ with the standard concentration $C_0 = 1$ mol/l is problematic, as recently pointed out by Hall, Dixon and Dickson (2020, ChemRxiv). We therefore refrain from doing such direct comparisons of our reaction-coordinate dependent free energies with standard free energies of binding, and but rather suggest a qualitative comparison as a rough consistency check. Instead, we explicitly calculate K_D values together with error bars from our kinetic data and compare them to experimentally obtained values in Fig. 2.

REVIEWERS' COMMENTS:

Reviewer #1 (Remarks to the Author):

I am happy with the changes made by the authors and gladly recommend publication .

- Pratyush Tiwary

Reviewer #3 (Remarks to the Author):

The authors satisfactorily addressed all my previous concerns. Therefore, I recommend the manuscript for publication in the present form